# Small Mammal Diversity Changes in a Baltic Country, 1975–2021: A Review

**DOI:** 10.3390/life12111887

**Published:** 2022-11-14

**Authors:** Linas Balčiauskas, Laima Balčiauskienė

**Affiliations:** Nature Research Centre, Akademijos 2, 08412 Vilnius, Lithuania

**Keywords:** small mammals, diversity trends, community structure, trophic groups, long-term

## Abstract

**Simple Summary:**

Mediated through functional and numerical responses, the community structure and diversity of small mammal communities over the long term may show the influences of climate change, landscape changes and local disturbances. We present an overview of small mammal trapping and owl pellet analysis results in pre- and post-soviet Lithuania (the most southerly of the three Baltic States, Northern Europe), covering the period 1975–2021. All available sources, including grey literature, were included where the raw data were available. Based on the decades-long period, we analysed diversity changes and proportions within the main trophic groups. Diversity increase and dominance decrease occurred in the aftermath of changes within the country in 1990 and have not subsequently changed. Thus, there are two periods—before and after large-scale agricultural changes in the country. The proportions of the trophic groups changed gradually: a large increase in granivores coincided with a decrease in omnivores and insectivores, while the proportion of small herbivorous mammals increased less notably.

**Abstract:**

The structure and diversity of small mammal (SM) communities over the long term may show the influences of climate change, landscape changes and local disturbances. We review published data regarding SM trapping and owl pellet analysis from Lithuania (the most southerly of the three Baltic States, Northern Europe), covering the period 1975–2021. Over decades, we analysed trends in the diversity of SM communities and the proportions of species and proportions of trophic groups. The large increase in granivores, from 6.9% in 1975–1980 to 45.4% in 2011–2020 and 54.7% in 2021, coincided with a decrease in omnivores and insectivores. The proportion of herbivores increased less notably. At the species level, significant decreases in the proportions of *M. arvalis*, *C. glareolus* and *S. araneus* were accompanied by notable increases in the proportions of *A. flavicollis* and *A. agrarius*, the latter from 1.0% in 1975–1980 to 25.3% in 2021. Concluding, two periods were identified, specifically before the 1990s and subsequently. In the second period, in the aftermath of land-use changes within the country in 1990, diversity increased, and dominance decreased, a situation that has not subsequently changed. Not excluding the concomitant effects of climate change, we relate these patterns to the alterations in habitat and anthropogenic impact.

## 1. Introduction

It is known that small mammal community structure and diversity over the longterm are affected by climate change [1], local disturbances [2,3] and landscape changes [4]. Small mammals are important for their capacity to respond to environmental and climatic changes through functional and numerical responses [5]. These changes are visible in small mammal trapping [6] and in changes in the diet of raptors and owls [7,8].

Knowledge of the response of small mammal communities and species is required to understand the future requirements and methods of biodiversity conservation and to meet upcoming challenges [9]. As shown by long-term small mammal monitoring [10], some species or groups are better suited to varying ecological, economical and epidemiological situations, therefore, are more resilient. Some small mammal species have shown long-term synchronised fluctuations, suggesting shared food resources, shared predators or both [6]. Small mammal monitoring is able to predict the sustainability and stability of populations, e.g., during a 46-year-long study in the Southern Yukon (60° N), the proportion of two of the main four small mammal species, *Myodes rutilus* and *Peromyscus maniculatus*, increased and decreased by 22% respectively [11]. Understanding the reasons for such changes requires more data to see perspectives that might otherwise be missed [12]. Short-term data might produce incorrect conclusions regarding the impact of vegetation succession on the small mammal community structure, species interactions and resilience [2].

Landscape, mainly the structure and composition of its mosaic, is reported as one of the main drivers of change in small mammal communities [12]. Similarly, changes in agricultural activities, namely the fallowing of land previously intensively used, can create new habitat sources for small mammals. As suggested by [13], this might lead to the extinction of some small mammal species and an increase in the number of yellow-necked mice (*Apodemus flavicollis*). At the same time, a decline in diversity in small mammal communities can occur [13].

In Central Europe, the findings of Zárybnická and co-authors revealed the significance of local biotic and abiotic factors affecting the structure and dynamics of small mammal communities. They noted that while the abundance of *M. glareolus* and *A. flavicollis* increased over time in the young forest, those of *S. araneus* decreased. By contrast, the community structure in the mature forest did not change significantly over time [14]. Based on this, the initial effect of restoring woodland in agricultural landscapes may be beneficial for SM, increasing their diversity [15]. Results of studying meadow-forest succession, however, have revealed that the diversity of the small mammal community decreases with increasing forest age, the small mammal communities being the most diverse in the re-growing meadows [16].

Recognising small mammals as a model group to evaluate the effects of climate change and shifts in land use [1], many investigations have been conducted in dry or desert environments [17,18,19,20]. These studies have highlighted the importance of habitat disturbances and cover layers [17], the delayed effects of forest fires [18] and the precipitation regime [18,19,20]. Mediation of the climate factor could be different [21], including snow cover depth and duration [22]. Information regarding the influencing factors and prediction of possible responses of small mammals could be found in [23,24].

The diets of mycophagous birds and other animals are closely related to the status of small mammal species and communities [7,8,25]. Therefore diet studies of the mentioned animals are used as a method to assess species composition and abundance of small mammals [8,9,26,27,28]. For example, a long-term study of the diet of the asp viper (*Vipera aspis*) confirmed an increase in *M. glareolus* and a decrease in *S. araneus* in the forest zone of Italy [25]. Likewise, based on the owl diet, a reduction in shrews was noted in Italy; these were replaced by rats *(Rattus norvegicus*) and house mice (*Mus domesticus*) [29]. In this case, only a slight variation in the richness and diversity of the small mammal community was observed. Therefore, we used data on the diet of the tawny owl (*Strix aluco*) and long-eared owl (*Asio otus*) from Lithuania to see for trends in the small mammal composition, but not to compare with the data of trappings.

The aims of this review were (i) to collect all available data on small mammal trapping and owl pellet analysis in Lithuania, the most southerly of the three Baltic states, and (ii) to analyse long-term changes in small mammal diversity and composition of their communities based on trophic groups. Based on the responses of small mammals to changes in land use [2,3,7,9,12,13,14,15,16], we tested the hypothesis that small mammal diversity increased after the 1990s when large-scale intensive agriculture was abandoned along with the other land use changes after the regaining of the country’s independence. This model may be representative of the other post-soviet Baltic countries.

## 2. Materials and Methods

### 2.1. Collection of the Published Material

Published results for the period of 1975–2021 were used for this study. We do not conform to PRISMA requirements for information retrieval, as (1) many of our published sources are not indexed in search systems, and (2) we used all available sources where the raw results concerning small mammal trapping or pellet analyses were available. If the same data were replicated in the later publications, earlier data were not included.

This publication is the first to use the full set of small mammal trapping and *S. aluco* pellet data in Lithuania. All published data are available in the mentioned sources [3,16,30,31,32,33,34,35,36,37,38,39,40,41,42,43,44,45,46,47,48,49,50,51,52,53,54,55,56,57,58,59,60,61,62,63,64,65,66,67,68,69,70,71,72,73,74,75,76,77,78]. Further analyses will follow, and therefore the dataset is not deposited as a public source. Geo-referencing of data collection sites is not used in this paper, as we analyse only the general trend on small mammal diversity and species composition across the country using data pooled in decades.

### 2.2. Dataset

Data on small mammals trapped between 1975–2021 include 475 samples from 165 unique sampling sites with 376,788 trap-days. Data on small mammals retrieved from the owl pellets cover the period 1986–2009 and include 52 batches. Due to the differences between trapping results and *S. aluco* diet [58], both sets were not pooled. The species composition of small mammals is presented in Table 1. Species richness, however, might be dependent on methodological problems–*N. anomalus*, *A. sylvaticus* and *M. rossiaemeridionalis* in the pellets were not identified.

To assess temporal trend, we arbitrarily grouped the data into six periods: 1975–1980, 1981–1990, 1991–2000, 2001–2010, 2011–2020 and 2021. We consider the first five periods to be long enough to avoid possible bias in the trend estimation [79], while for 2021, we just assessed if the sampling effort is sufficient.

In this paper, we do not intend to analyse habitat-based or site-based patterns of small mammal diversity. Therefore, the sampling effort is presented at the level of the main groups of habitats and shows the regions of the country where the small mammals were trapped (Table 2). The main investigated habitats were forests (all types and ages), wetlands (marshes, bogs, swamps, raised bogs and peatbogs), meadows (all types of meadows and pastures), habitat complexes (including forest, wetland and meadow habitats in about equal proportions), and other habitats (including agricultural fields, orchards, farmsteads, river banks, lakeshores and islands). 

Investigations were not equal across all of the main habitats (Table 2), except in the 1981–2000 period when the main habitats (forests, wetlands and meadows) were investigated with similar effort (22,610, 15,920 and 16,300 trap-days, respectively). In the 1975–1980 period, the highest trapping effort was in meadows (13,450 trap-days), while wetlands were under-trapped from 2001 (6349 trap-days), and more attention was paid to the habitats other than three main ones from 2011, these including orchards (21,228 trap-days), cormorant colonies (15,475 trap-days), farmsteads and other buildings (4960 trap-days), city parks, etc. For each decade, we also mention the aims of the main small mammal investigation works and provide references for the published data sources. The best periods of trapping coverage in the country were related to research for a mammal atlas [34,36,37,42,46] and to an inventory of protected and other potentially biodiversity-rich territories [34,35,38,39,40,43,44,53,54,64], as well as during small mammal monitoring [48] and investigations of commercial orchards [72,74,75,76,77].

Studies of owl diet [49,50,51,56,57,58,61,62] were limited to the three periods within the years 1987 to 2009 and were not related to habitats (in most cases, forest meadows and their ecotones with agricultural land, these being within the owl hunting distances).

### 2.3. Data Analysis

To avoid the influence of non-equal sampling effort, we analysed species accumulation curves as well as diversity patterns using rarefaction based on the individuals [80,81], estimating how many species could be expected in a sample with a smaller total number of individuals. Combinatorial terms are computed as recommended in [82], using a log Gamma function and calculated in PAST version 4.01 (Paleontological Museum, University of Oslo, Oslo, Norway). Standard errors are given by the program and converted to 95% CI in the graphical plot. Rarefaction eliminates the influence of unequal trapping effort across the analysed periods. As in the previous analysis [34], we found a direct relation between trapping effort and the number of trapped small mammal individuals and, to a lesser extent, the number of registered species. As we refer to previous studies, this shortcoming is unavoidable.

The analysed indices of the small mammal communities were an unbiased number of species, dominance (D) and diversity (Shannon’s H) [82]. According to Ref. [82], dominance ranges from 0 (all species are equally present in the community) to 1 (one taxon dominates completely). Differences between the above-mentioned indices across the compared periods and the upper and lower limits of D and H were calculated using bootstrap with *n* = 9999, while for the number of species, Chao-1 was used. In PAST, the given number of random samples was produced, each with the same total number of individuals as in the original sample. For each individual in the random sample, the taxon was chosen with probabilities proportional to the original abundances. Then, 95% CI was calculated. The proportions and the 95% CI for the four trophic groups among all trapped small mammals were calculated with the Wilson method of the score interval [83] using OpenEpi epidemiological software [84]. Differences in the proportions of the most abundant species between periods were evaluated using the G test with an online calculator [85]. 

## 3. Results

From 1975 to 2021, the dominant species of small mammals in Lithuania was *C. glareolus*, with a proportion of 38.0% (95% CI = 37.6–38.5%) among all trapped individuals, followed by *A. flavicollis* with 17.5% (17.2–17.9%), *A. agrarius*, 10.5% (10.2–10.8%), and *S. araneus* with 10.0% (9.8–10.3%). Species with proportions 1 to 10% were *M. arvalis*, 9.2%, *M. oeconomus*, 5.2%, *M. agrestis*, 2.9%, *S. minutus*, 3.3%, *M. minutus*, 1.1% and *M. musculus*, 1.1%. The proportions of the remaining small mammal species were less than 1% each. Differences in the proportions of all listed species are significant (G-test, *p* < 0.001). Species richness, the dominance index D and diversity, as Shannon’s index, are presented in Table 1. These parameters had expressed trends in the study period.

The dominant species of the owl that hunted small mammals from 1986 to 2009 was also *C. glareolus*, with a proportion of 29.5% (95% CI = 28.36–30.7%). It was followed by *A. flavicollis* with 19.8% (18.8–20.9%), *M. arvalis*, 12.6% (11.7–13.4%), *M. agrestis*, 11.8% (11.0–12.7%) and *S. araneus* with 11.7% (10.9–12.6%). Species with proportions 1 to 10% were *M. oeconomus*, 2.7%, *S. minutus*, 2.6%, *A. agrarius*, 2.5%, *M. avellanarius*, 2.1% and *A. amphibius*, 1.1%. Differences in the proportions of all listed species are significant (G-test, *p* < 0.001). 

Due to differences in small mammal diversity (*t* = 10.4, *p* < 0.001) and dominance (*t* = 13.1, *p* < 0.001), these two data sets will be analysed separately. As mentioned in the Section 2.2, three small mammal species in the pellets were not identified.

### 3.1. Changes in Trapped Small Mammal Diversity and Species Richness in Lithuania, 1975–2021

Small mammal diversity rose in the 1990s and thereafter remained at this higher level until the current decade, with the dominance index showing the opposite trend, decreasing in the 1990s (Table 3). Therefore, there are two distinct periods: the period prior to the 1990s, characterised by lower diversity and the period after the 1990s, with diversity increase. These differences are statistically significant.

Diversity indices in 2021 were different from the other decades. However, this period was characterised by very limited trapping effort and biased habitat proportions (Table 2). The above-mentioned trend was not related to different sampling efforts. Species accumulation curves show that a sample of over 1000 individuals is required to reach 14–18 species (Figure 1a). Thus, the results of the latest decade (Table 3), presented so far by trappings of 2021, should improve on further sampling. To get a full representation of the diversity index, however, the sample size does not need to be over 1000 trapped individuals (Figure 1b), as Shannon’s H remains at the same level with the increased sample. Therefore, differences in small mammal diversity are unbiased by a different number of trapped individuals in the compared periods.

Dominance in the first period, 1975–1980, depended on the high proportions of *M. arvalis* and *C. glareolus* among the trapped SM, these being 44.0% (CI = 42.4–45.6%) and 35.0% (CI = 33.4–36.5%), respectively, despite trapping effort was mainly targeted to meadows (Figure 2a). From this period onward, the proportion of *M. arvalis* decreased to 4.7%, 6.6%, 6.3%, 7.5% and 10.0%, respectively, the trend being highly significant (G = 10,321, *p* < 0.0001). In the subsequent decades, the proportion of *C. glareolus* was 63.4%. 42.6%, 36.1%, 26.3% and 25.2%, respectively. Thus, the decreasing trend is highly significant (G = 37,600, *p* < 0.0001) and not related to the trapping effort in forest habitats (Table 2). A decrease in the proportion of *S. araneus* was less visible, this falling from 12.4% in 1981–1990 to 8.6% of all trapped small mammals in 2011–2020 (G = 55.3, *p* < 0.0001).

These decreases were accompanied by notable increases in the proportions of *A. agrarius* and *A. flavicollis*, both trends being significant (G = 2227 and G = 1302, respectively, *p* > 0.0001). The proportion of *A. agrarius* rose continuously over time, from 1.0% (CI = 0.7–1.4%) in 1975–1980 to 25.3% (22.2–28.8%) in 2021, with the respective proportions for *A. flavicollis* being 5.5% (4.8–6.3%) and 29.1% (25.8–32.5%). Changes in proportions of the other species were not so distinct (Figure 2b) as the proportions of this group accounted for only 0.3–1.7% of all trapped SM (Figure 2a).

### 3.2. Changes in the Proportions of Trapped Small Mammal Trophic Groups

Based on the changes in species, trends in the representation of the functional groups after 1980 in the small mammal community are even more visible (Figure 3). Growth in the proportion of granivores was from 6.9% in 1975–1980 to 45.4% in 2011–2020 and 54.7% in 2021 (G = 8837, *p* < 0.0001), occurring in concert with a shrinking proportion of omnivores, from 63.6% in 1981–1990 to 26.8% in 2011–2020 and 28.6% in 2021 (G = 2249, *p* < 0.0001). An increase in the proportion of herbivores from 8.8% in 1981–1990 to 18.0% in 2001–2010, 15.9% in 2011–2020 and 13.6% in 2021 was also significant (G = 265.0, *p* < 0.0001). The decrease in the proportion of insectivores in 1981–2020 was less expressed (G = 181.0, *p* < 0.001).

### 3.3. Diversity of Small Mammals Recovered from Owl Pellets

Despite highly unequal sample sizes, the diversity and dominance of the small mammals in the owl pellets were stable across the three periods from 1986 to 2009. Species richness, however, significantly increased along with the sample size (Table 4).

With both datasets originating from the same region and mostly from the same locations, the composition of the prey in 1991–2000 and in 2001–2010 was stable, both in terms of species (Figure 4a) and trophic group (Figure 4b). Differences in the proportions of *S. araneus* (G = 2.7). *A. flavicollis* (G = 2.07), *M. arvalis* (G = 0.28), *C. glareolus* (G = 0.12), insectivores (G = 3.7), herbivores (G = 2.7), omnivores (G = 0.57) and granivores (G = 0.50) were not significant (*p* > 0.07).

The earliest sample was characterised by a higher proportion of *M. avellanarius* and the synanthropic small mammal species *R. norvegicus* and *M. musculus* (Figure 4a), thus yielding an over-representation of omnivores and an under-representation of insectivores (Figure 4b).

## 4. Discussion

The hypothesis we tested was confirmed; namely, small mammal diversity increased after the 1990s. This was in line with a large decrease in the proportion of small omnivorous mammals, a decrease in insectivores, a large increase in granivores and an increase in herbivores. 

In Lithuania, two distinct periods of recent land use can be identified: (i) 1960–1990, characterised by massive land reclamation and unvaried Soviet agriculture; (ii) after 1990, characterised by rapid changes in land ownership, increased fragmentation, fallowing of former agricultural areas, increase in the intensity of forest use and development of the small-scale forestry [86,87]. Abandonment of unproductive land was followed by increased forest cover, urban development and a decrease in crop area [88]. In Lithuania, the abandonment rate of agricultural land between 1990 and 2000 was 28% [89]. Changes in the usage of herbicides were followed by an increase in segetal flora, i.e., weeds in the crops [90].

In 1971, three main habitats, namely forest, agricultural land and meadows/pasture, covered equal proportions of land in Lithuania, around 28–30% each, with other land used accounting for less than 12% [91]. This proportion was retained in trapping efforts up to 1990 (see Table 2). From 1990 to 2018, the main land cover changes were: a decrease in heterogeneous agricultural areas by 2315 sq. km and pastures by 361 sq. km, an increase in scrub and herbaceous vegetation areas by 1485 sq. km, in arable land by 816 sq. km and in forest areas by 244 sq. km [92]. While trapping effort in the 1991–2010 period was comparable to habitat structure (see Table 2), habitat changes were not reflected. Low wetland representation in trapping effort after 2011 do not fully coincide with the moderate decrease of inland wetlands after 1990 [92].

Owls were able to bypass land use changes, as the preyed-upon habitat proportions around the breeding areas of *S. aluco* between 1995–2004 and 2005–2014 were similar [93]. Still, however, there was a sharp decrease in herbivores, mainly voles of the g. *Microtus*, along with relative stability of murids (granivores) as well as *C. glareolus* (omnivore) proportions in the prey in the periods of 1978–1989, 1990–2001 and 2002–2014 [94]. However, the absence of raw data from these time periods did not allow us to incorporate the dataset into our analysis. 

It is commonly accepted that land use changes have aftereffects on small mammal communities; see references in [95,96]. In some cases, species richness and composition might not be affected by the intensity of agriculture, though the species assemblages in low, medium and highly intensified landscapes might be different [7]. Agriculture should favour habitat generalists, such as *C. glareolus*, but an increase of this species in boreal landscapes [97] contradicts our finding in the middle latitudes. Further south, in Italy, under agricultural development, insectivores were replaced by commensal generalist species [29]. These authors exclude the influence of climate change and broad-scale land use from the list of factors that drove small mammal community composition.

A decline in insectivores in agricultural lands can be related to increased grain crops and the spread of herbicides and insecticides, both reducing the diversity and abundance of invertebrate prey [29]. As the intensification of agriculture negatively affects ecosystem services, all natural and semi-natural patches become very important for small mammals in the agricultural matrix. An increase in permanent crops was shown to be advantageous for *A. agrarius* [98]. Likewise, supplemented with variables of temperature, the proportions of the cover of crop type could be used to forecast *A. agrarius* outbreaks in China agroecosystems [99].

Therefore, we need to recognise that small mammal species differ in their responses to changes in landscape or landuse and that habitat quality or composition is no less important than patch size, fragmentation and isolation [100]. Mosaics of disturbed land, including both agricultural and pastoral ones, provide important habitats for diverse species of small mammals [101]. When land patches undergo succession, small mammals may be affected by both patch size variations and successional changes [102]. For example, during meadow-forest succession in Lithuania, small mammal diversity declined as typical meadow species were replaced by a few forest small mammal species [16].

Growth in the afforested area of Lithuania has been characteristic to recent decades, with the increase since the 1990s being 3.4% [103], and the current forest cover 33.7% of the entire country area [104]. Forest habitat decreases the diversity of SM, especially when forest age advances [16]. At the species level, changes in dominant species occur, from *Microtus* voles to *Clethrionomys* [105]. From [106,107], we might expect further decreases in the diversity of small mammals in forests as minimum diversities are achieved when the forest stands are about 25–40 years of age, prior to clearcutting. Richer small mammal communities in younger stands are also characteristic of Central Europe [14]. In line with land fallowing [13], considerable declines in small mammal species diversity should be expected.

The impact of global climate change on small mammal populations is widely recognised in various latitudes, from the northern taiga [11,108] and the temperate zone and middle latitudes, see [25] and references herein, to the tropical zone [109,110]. Factors related to climate change, such as reduction of precipitation, droughts and other extreme events, cause additional damage to the anthropogenic pressures [111]. In this review, however, we do not focus on the effect of climate change. Following [25], local factors, such as the composition of habitats and land use changes, were given priority.

The impact of land-use change on small mammal communities can vary geographically. In Hungary, for example, grassland restoration had little local impact on small mammal communities, while habitat management was important [112]. In contrast, in the mid-west region of North America, restored native grasslands have shown the importance of restored areas, acting as survival stations for voles at a time of population decline in agricultural landscapes [113]. Therefore, the synthesis of analyses of long-term changes in small mammal communities will be of great predictive value. Following Ref. [114], further evolution of small mammal investigations in Lithuania should proceed in two directions: (i) monitoring in selected areas of the main habitats to see the direction of long-term changes, and (ii) investigations of previously non-existent or not covered habitats, such as biofuel plantations, restored peatlands, afforested quarries, etc.

## 5. Conclusions

Analysing long-term small mammal trapping from Lithuania (Northern Europe), we found two periods with different compositions of communities—before the 1990s and after. The second period is characterised by land- and forest-use changes related to regained independence from the Soviet Union. The former period was characterised by lower diversity and higher dominance. In the second period, the proportion of omnivorous and insectivorous species decreased; these groups were replaced by increasing granivorous and herbivorous species.

The stability of diversity indices after 1990 occurred despite uneven trapping efforts in the main habitats (forests, meadows and wetlands). Not excluding the concomitant effects of climate change, we relate these changing patterns of small mammal diversity to the alterations in land use and anthropogenic impact. 

## Figures and Tables

**Figure 1 life-12-01887-f001:**
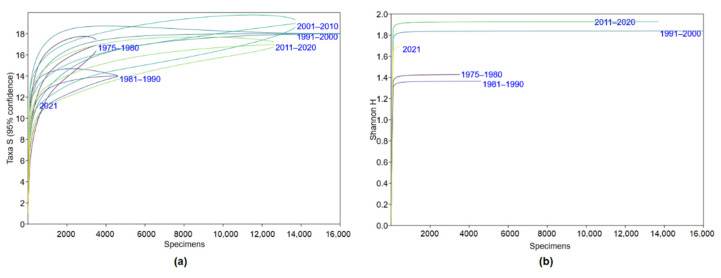
Analysis of the dependence of diversity trends on the sampling effort in small mammal trapping: (**a**)—species accumulation curves, (**b**)—the pattern of diversity index, H.

**Figure 2 life-12-01887-f002:**
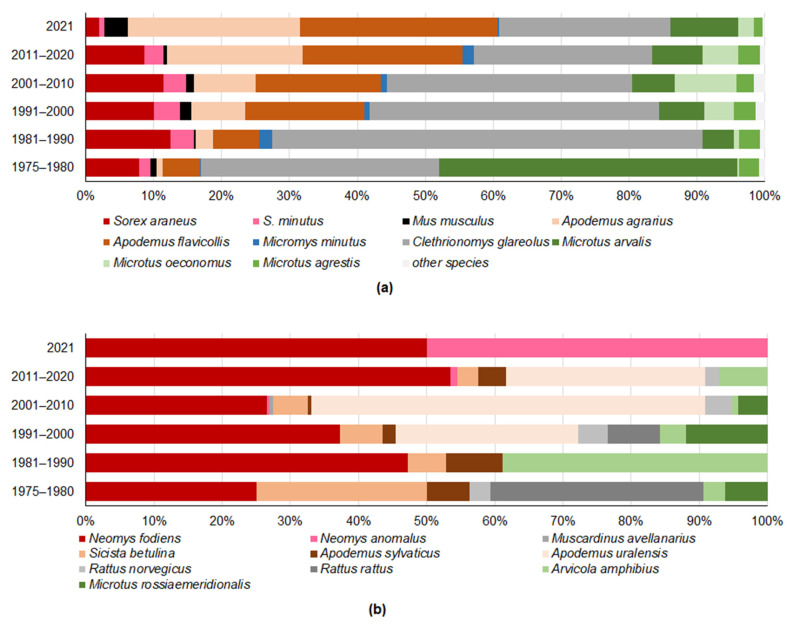
The trend in the proportion of trapped small mammal species in Lithuania by the decade 1975–2021: (**a**)—species with over 1% of all trapped individuals, (**b**)—other species (with under 1% of all trapped individuals).

**Figure 3 life-12-01887-f003:**
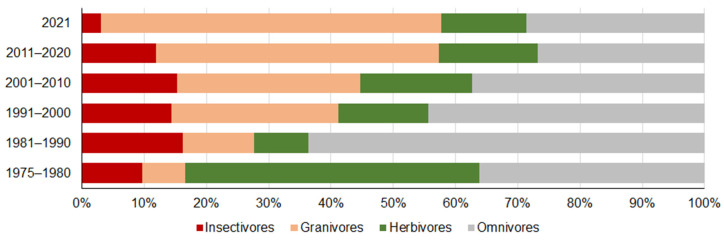
Trends in proportions of small mammal trophic groups in Lithuania by decade, 1975–2021.

**Figure 4 life-12-01887-f004:**
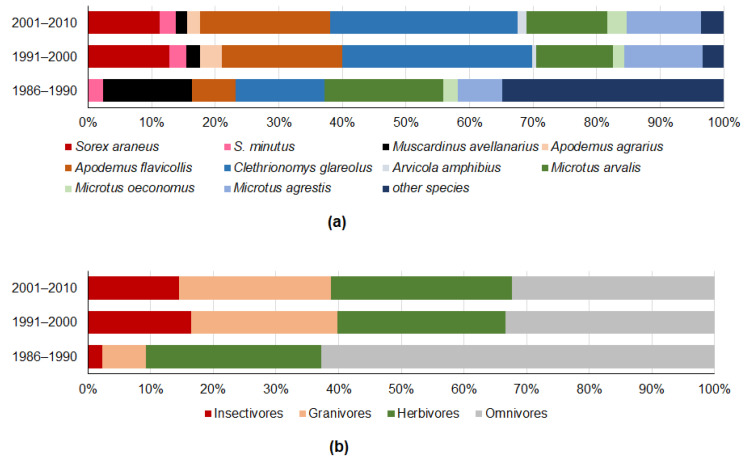
Proportions of small mammal species recovered from the owl pellets (**a**) and proportions of trophic groups (**b**) by decade.

**Table 1 life-12-01887-t001:** Species composition of small mammals trapped in Lithuania between 1975–2021 and retrieved from owl pellets in the period 1986–2009. Trophic groups: I—insectivores, G—granivores, H—herbivores, O—omnivores, according to Ref. [71].

Species	Trophic Group	Trapping, Ind.	Pellets, Ind.	Total, Ind.
Common shrew (*Sorex araneus*)	I	5182	676	5858
Pygmy shrew (*S. minutus*)	I	1695	149	1844
Water shrew (*Neomys fodiens*)	I	218	49	267
Mediterranean water shrew (*N. anomalus*)	I	3	0	3
Hazel dormouse (*Muscardinus avellanarius*)	O	1	119	120
Northern birch mouse (*Sicista betulina*)	G	38	53	91
House mouse (*Mus musculus*)	O	544	42	586
Striped field mouse (*Apodemus agrarius*)	G	5434	142	5576
Yellow-necked mouse (*A. flavicollis*)	G	9079	1145	10,224
Wood mouse (*A. sylvaticus*)	G	14	0	14
Pygmy field mouse (*A. uralensis*)	G	218	1	219
Harvest mouse (*Micromys minutus*)	G	582	36	618
Brown rat (*Rattus norvegicus*)	O	21	25	46
Black rat (*R. rattus*)	O	26	0	26
Bank vole (*Clethrionomys glareolus*)	O	19,673	1703	21,376
Water vole (*Arvicola amphibius*)	H	32	63	95
Common vole (*Microtus arvalis*)	H	4782	725	5507
Root vole (*M. oeconomus*)	H	2693	154	2847
Short-tailed vole (*M. agrestis*)	H	1523	683	2206
Sibling vole (*M. rossiaemeridionalis*)	H	37	0	37
Total, N		51,795	5775	57,570
Species, S		20	16	20
Diversity, H		1.90	2.03	1.93
Dominance, D		0.21	0.17	0.20

**Table 2 life-12-01887-t002:** Representativeness and genesis of small mammals trapping in Lithuania between 1975–2021.

Period	Map	Trapping Purpose	Trapping Effort			References
			TS	TD	By Habitat	
1975–1980	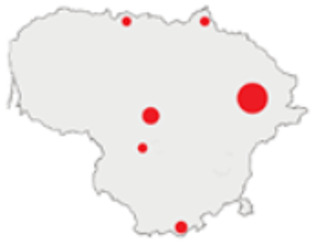	IM, IPA, M, P	6	16,330	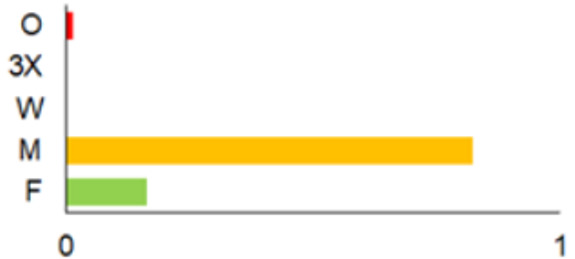	[31,32,33]
1981–1990	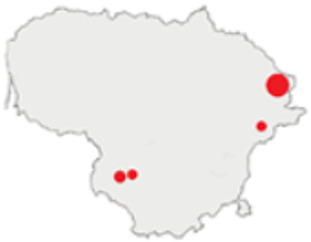	INPP, IPA, ODI, P	4	50,250	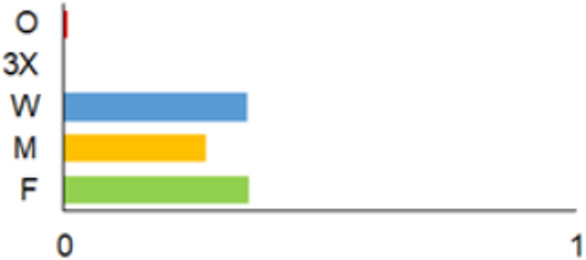	[34,48,49]
1991–2000	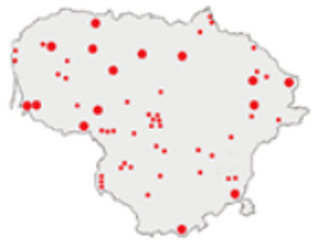	IPA, IBT, IFS, IS, SMM, M, MA, ODI	69	86,165	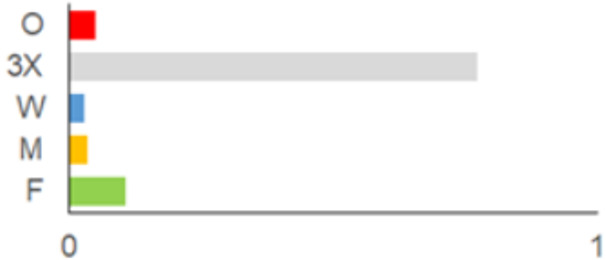	[35,36,37,38,39,40,41,42,43,44,45,49,50,51]
2001–2010	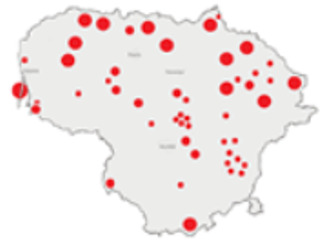	IFS, IPA; SMM, M, ODI, IFM, IS	35	116,442	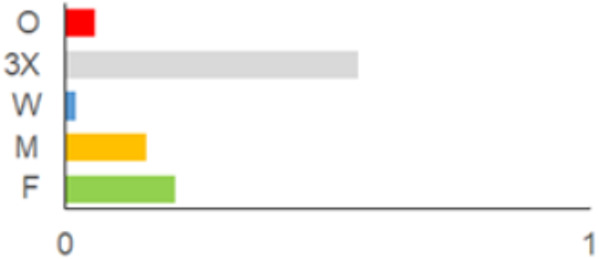	[46,47,49,50,51,52,53,54,55,56,57,58,59,60,61,62,63]
2011–2020	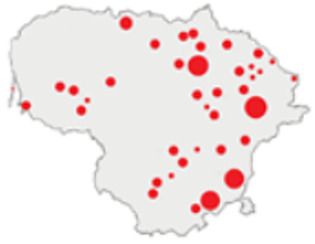	ICC, ICH, IFM, IO, P, M	40	98,693	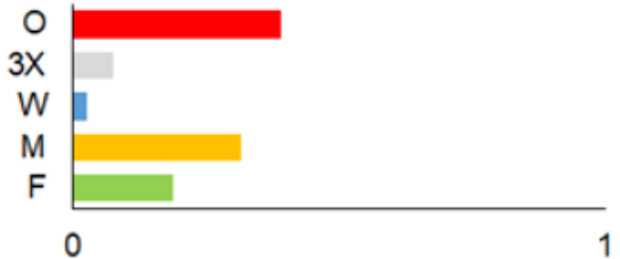	[3,16,64,65,66,67,68,69,70,71,72,73,78]
2021	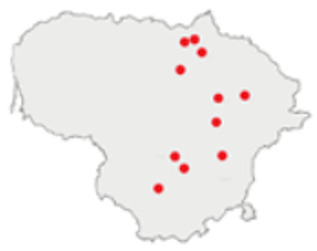	ICC, ICH, IO, P	11	8908	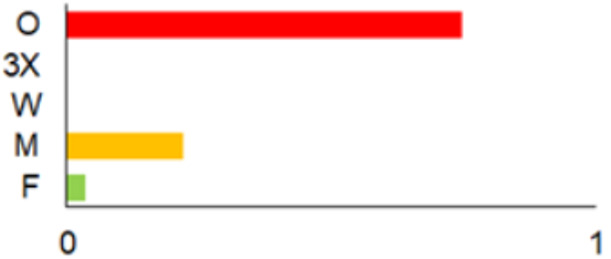	[74,75,76,77]

Trapping purpose: IPA—inventory of protected areas; P—material for parasitological research; INPP—small mammal monitoring in the region of Ignalina Nuclear Power Plant; IBT—inventory of potentially biodiversity-rich territories; SMM—small mammal monitoring program, state level; IO—investigations in commercial orchards; ICH—investigation of commensal habitats; ICC—investigation of colonies of great cormorants (*Phalacrocorax carbo*); IFS—investigation of meadow-forest succession; IM—inventory of meadows; IS—inventory of islands; M—miscellaneous; MA—mammal atlas works; ODI—owl diet inventory, IFM—Investigation of flooded meadows. TS—number of unique trapping sites, TD—trap-days. Main trapping habitats: F—forests; M—meadows; W—wetlands; 3X—forests + wetlands + meadows combined; O—other, presented as proportions (0 to 1).

**Table 3 life-12-01887-t003:** Trends of trapped small mammal diversity indices in Lithuania by decade, 1975–2021. For the number of species, the limits of Chao-1 estimates are given in parentheses. For diversity and dominance indices, bootstrap values are given. Different superscript letters denote significant differences at *p* < 0.001.

Index	1975–1980	1981–1990	1991–2000	2001–2010	2011–2020	2021
Individuals, N	3637	4729	16,144	13,854	12,764	667
Number of species, S	17 (17–23)	14 (14–15)	18 (18–18)	19 (19–25)	17 (17–20)	12 (12–15)
Diversity, H	1.43 ^a^ (1.39–1.46)	1.36 ^b^ (1.33–1.40)	1.84 ^c^ (1.82–1.86)	1.93 ^d^ (1.91–1.94)	1.93 ^d^ (1.91–1.94)	1.69 ^e^ (1.63–1.75)
Dominance, D	0.33 ^a^ (0.32–0.34)	0.43 ^b^ (0.41–0.44)	0.24 ^c^ (0.23–0.24)	0.20 ^d^ (0.20–0.20)	0.18 ^e^ (0.18–0.18)	0.22 ^c^ (0.21–0.24)

**Table 4 life-12-01887-t004:** Diversity indices of small mammals recovered from owl pellets in Lithuania by decade, 1986–2009. For the number of species, the limits of Chao-1 estimates are given in parentheses. For diversity and dominance indices, bootstrap values are given. Different superscript letters denote significant differences at *p* < 0.001.

Index	1981–1990	1991–2000	2001–2009
Individuals, N	43	2029	3703
Number of species, S	9 (8–12) ^a^	16 (16–16) ^b^	15 (15–15) ^b^
Diversity, H	2.02 ^a^ (1.81–2.09)	2.02 ^a^ (1.97–2.05)	2.03 ^a^ (2.00–2.06)
Dominance, D	0.15 ^a^ (0.14–0.20)	0.17 ^a^ (0.17–0.18)	0.17 ^a^ (0.17–0.18)

## Data Availability

This is ongoing research; therefore, unpublished data are not available publicly. All other data are available in the cited publications.

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
