# Peer review of "Small Mammal Diversity Changes in a Baltic Country, 1975–2021: A Review"

_life, 2022, doi:10.3390/life12111887_

Round 1

Reviewer 1 Report

The manuscript was well-written. It contains comprehensive information about mammals diversity of Lithuania. It would be better to reword the manuscript's title. It would also be better to improve the abstract and summary. An abstract or a summary should contain at least a concise background sentence, justification of the manuscript or research, important/interesting findings or discoveries and a conclusive, if applicable.

If possible, selected images and scientific data of important small mammal species (globally threatened or endemic species) recorded in the country over research by the author should be included in a review for more attractive and interesting.

Author Response

Rev#1 comments and answers

Comment: The manuscript was well-written. It contains comprehensive information about mammals diversity of Lithuania. It would be better to reword the manuscript's title. It would also be better to improve the abstract and summary. An abstract or a summary should contain at least a concise background sentence, justification of the manuscript or research, important/interesting findings or discoveries and a conclusive, if applicable.

Answer: to acknowledge your comment. we change title as „Small mammal diversity changes in a Baltic country, 1975–2021: a review“. We also extended summary with introductory sentence: „Mediated through functional and numerical responses, the community structure and diversity of small mammal communities over the long-term may show the influences of climate change, landscape changes and local disturbances.“ We rewrote Abstract, adding concluding remark and introductory sentence.

Comment: If possible, selected images and scientific data of important small mammal species (globally threatened or endemic species) recorded in the country over research by the author should be included in a review for more attractive and interesting.

Answer: unfortunately, in Lithuania we do not have any small mammal species being globally threatened or endemic, therefore there are no images you expect to be included into the paper.

Reviewer 2 Report

The authors address an important quesion, the changes in small mammal diversities in Lithuania, using a large, highly valuable dataset. The ms is well written, the text needs a moderate English editing. Although I am happy with the structure of Introduction and Discussion, the Methods need clarification at several points, as detailed below.

When analysing temporal trends in the species richness, dominance and diversity, I suggest to classify the dataset into 5-years intervals and fit a linear regression on the response variables as a result of years. If the records are spatially clumped, I suggest to apply a linear mixed model, but analysing spatial autocorrelation (e.g. calculating Moran’s I) is definitely necessary.

Materials and Methods:

Line 108: How many unique location did you have?

Line111: provide a reference for classifying the specimens into trophic groups

Table 1: add the number of unique locations to the table

Line 117: Why did you choose this approach to create arbitrary temporal periods?

Lines 129-133: provide data on the variance in effort

Line 158: provide details for the rarefaction

Line 164: define dominance and add reference

Line 167: which bootstrap statistic did you use?

Line 171: provide reasons why did you apply G-test

Author Response

Rev#2 comments and answers

Comment: The authors address an important quesion, the changes in small mammal diversities in Lithuania, using a large, highly valuable dataset. The ms is well written, the text needs a moderate English editing. Although I am happy with the structure of Introduction and Discussion, the Methods need clarification at several points, as detailed below.

Answer: thank you for the positive comments. Language of the paper was edited by professional editor and native speaker, but we might discuss about the text changes you propose. Answers to other comments, point to point, are presented below.

Comment: When analysing temporal trends in the species richness, dominance and diversity, I suggest to classify the dataset into 5-years intervals and fit a linear regression on the response variables as a result of years. If the records are spatially clumped, I suggest to apply a linear mixed model, but analysing spatial autocorrelation (e.g. calculating Moran’s I) is definitely necessary.

Answer: We thank you for this comment, however, please note this is review based on the published data with analysis limited to calculations of diversity and proportions, therefore statistics was limited to evaluation of the significance of differences. Using 5 year periods would yield even higher variances of the trapping effort, and, as some data were published not by year, but by periods, several good samples cannot be grouped in the shorter periods. Raw data were not available from the cited publications.

Analysis with response variables, such as changes in forest area, agriculture intensity and climate parameters is foreseen in the other paper, however, we are going to try model of regime shifts in the presence of uncertainty about the qualitative and quantitative characteristics of the system under study. We mean, that influence of the listed factors on the small mammal community and separate species is not well defined and unidirectional, therefore uncertainty should be addressed in models other than linear or linear mixed. Again, calculating of the Moran’s I require data not present in the publications, but we are working towards geo-referencing of all sites where possible. Therefore, your comment will be considered in the future analyses.

Materials and Methods:

Comment: Line 108: How many unique location did you have?

Comment: Table 1: add the number of unique locations to the table

Answer: please note, that number of unique locations in some trapping sites are quite big, 47, 38, 17 etc. – however, data cannot be traced to evely location. This is review, not all data are collected by authors, and not all raw data are traceable. Therefore, we acknowledge your comment by providing number of sites in the Table 2, and this number is 165 in total. It is inserted into the text before Table 1.

Comment: Line111: provide a reference for classifying the specimens into trophic groups

Answer: closest reference to trophic groups of small mammal species of Lithuania is our paper of the resource partitioning, reference added to the Table 1 caption.

Balčiauskas, L.; Skipitytė, R.; Balčiauskienė, L.; Jasiulionis, M. Resource partitioning confirmed by isotopic signatures allows small mammals to share seasonally flooded meadows. Ecol. Evol. 2019, 9, 5479–5489. https://doi.org/10.1002/ece3.5144

Comment: Line 117: Why did you choose this approach to create arbitrary temporal periods?

Answer: there are several reasons for using decade-long periods and their start-end years:

  1. Decade is the most widely used period in analysis of the medium-long time span (up to100 years); e.g., UN decades of ecosystems; strategic paper in Frontiers of ecology and the environment, DOI:10.1002/fee.2320, and others;
  2. We used 1-to-0 decade: in 1990 Lithuania regained independence, and 1991 was the first year extensive land use changes occurred
  3. 10 year period is long enough for changes in living nature (our case, small mammal communities) to take place
  4. Finally, 10 year periods had more balances trapping effort (less variance of it).

Comment: Lines 129-133: provide data on the variance in effort

Answer: requested numbers inserted into the text.

Comment: Line 158: provide details for the rarefaction

Answer: text changed as follows, adding two references:

“To avoid the influence of non-equal sampling effort, we analysed species accumulation curves as well as diversity pattern using rarefaction based on the individuals [80,81], estimating, how many species could be expected in a sample with a smaller total number of individuals. Combinatorial terms are computed as recommended in [82], using a log Gamma function and calculated in PAST version 4.01 (Paleontological Museum, University of Oslo, Oslo, Norway). Standard errors are given by the program and converted to 95% CI in the graphical plot. Rarefaction eliminates the influence of unequal trapping effort across the analyzed periods. As in previous analysis [34], we found a direct relation between trapping effort and the number of trapped small mammal individuals and, to a lesser extent, the number of registered species. As we refer to previous studies, this shortcoming is unavoidable.”

Comment: Line 164: define dominance and add reference

Answer: text added:

According [82], dominance ranges from 0 (all species are equally present in the community) to 1 (one taxon dominates completely).

Comment: Line 167: which bootstrap statistic did you use?

Answer: text added:

In PAST, the given number of random samples was produced, each with the same total number of individuals as in the original sample. For each individual in the random sample, the taxon is chosen with probabilities proportional to the original abundances. Then, 95% CI was calculated.

Comment: Line 171: provide reasons why did you apply G-test

Answer: G-test is used to evaluate significance of differences in proportions. One of our papers was continuously rejected for the reason not using this test for proportions... from the several reviews we understood it is a must.

Reviewer 3 Report

The authors present a neat study, nicely describing the trends and variation of the Lithuanian small mammal community. The manuscript would benefit from some reorganization and corrections.

I would suggest the authors to exclude 2021 from the study. Despite the reasoning on page 6 lines 204-213, I would suggest to drop the year 2021, from the manuscript. If the last decade would contain 3-4 years of sampling, it would be better. The five periods, without 2021, should be enough. Therefore I suggest reanalyzing the data.

Introduction: Based on what earlier knowledge did you make your hypothesis? So far it is not clear from the Introduction.

Some smaller remarks:

Pg 2, line 60: „Zárybnická and co-authors” number of publication missing.

Pg 2, line 60-68: Here I would suggest including the effects of grassland restoration and its management on small mammals, see Mérő et al. 2015, Animal Conservation.

Pg 2, line 71: delete „see references in”

Pg 6, lines 189-192: This part sounds more like methods than Results.

Pg 6, lines 206-207:Sentence for the Methods. In fact the part from lines 204-213, is more for the Methods part.

Pg 8, line 236, the phrase „other species” sounds awkward, I suggest to write „species with lower than 1% of all trapped individuals”. Please change this in the figures too.

Pg 10, lines 275-277, this sentence is redundant.

Pg 10, lines 278-293, this part is not discussion. This should be put in the Introduction or Methods. Most probably it is suitable in the Methods, under Study area subsection.

Author Response

Comment: Pg 2, line 71: delete „see references in”

Answer: deleted as advised.

Comment: Pg 6, lines 189-192: This part sounds more like methods than Results.

Answer: as advised, places in 2.2 chapter.

Comment: Pg 6, lines 206-207: Sentence for the Methods. In fact the part from lines 204-213, is more for the Methods part.

Answer: thank you, wording of the sentence was really as required for Methods part. We made change of this sentence, presenting it as result of analysis. In the Methods, necessary mentioning in now in beginning of 2.3 section.

Comment: Pg 8, line 236, the phrase „other species” sounds awkward, I suggest to write „species with lower than 1% of all trapped individuals”. Please change this in the figures too.

Answer: we acknowledge your comment, explaining what are “other species” in the caption of Figure 2. However, “other species” in the legend of Figure 2a seems acceptable, we consulted with the similar publications on the representation of species share.

Comment: Pg 10, lines 275-277, this sentence is redundant.

Answer: sentence deleted

Comment: Pg 10, lines 278-293, this part is not discussion. This should be put in the Introduction or Methods. Most probably it is suitable in the Methods, under Study area subsection.

Answer:

Review is based on the papers about trapping small mammals and owl diet, but not the habitats. In the former version we were recommended include mentioned information as the part of Discussion, to support findings about the changes of diversity and community structure.

In the Introduction we present a limited set of reference on the general patterns of small mammal diversity under different land-uses. In discussion, we present just a general data on the habitat changes during the analysed period, mostly referring to changes after 1990’s, the period with expressed changes in small mammals. Your comment will be acknowledged in full, when we prepare research article on the habitat preferences of separate small mammal species, and the analysis of habitat-related community patterns.

Round 2

Reviewer 2 Report

As all of my suggestions and concerns have been adequately addressed, I recommend to accept the article after a moderate English editing.

Reviewer 3 Report

As far as I see, the authors didn't consider my two major comments in my previous review report: did not find any responses and did not find changes in the manuscript. Please improve and clarify.

The responses on my smaller comments I can accept.

Author Response

Dear Rev#3,

we were under the impression that the system was missing part of our answer, so there is a reason for your frustration. Could you please check whether this answer has been included and whether it is acceptable?

This time I include pdf file of full answer to your comments. Our apologies, if relly part of the answer was missing.

Rev#3 comments and answers

Comment: I would suggest the authors to exclude 2021 from the study. Despite the reasoning on page 6 lines 204-213, I would suggest to drop the year 2021, from the manuscript. If the last decade would contain 3-4 years of sampling, it would be better. The five periods, without 2021, should be enough. Therefore I suggest reanalyzing the data.

Answer: we would like to argue with this comment. In the earlier version of the manuscript, last decade included 2021 and 2022 data, where trapping effort in 2022 was very limited, winter and early spring only. We were advised to exclude 2022 and keep 2021 data to assess sufficiency of the trapping effort.

Excluding 2021 did not change our conclusions, and the trends of changes in species and trophic groups remain the same. Significance did not suffer – yes, graphs would be shorter by one period. However, we will have no period with small trapping effort, and then credibility of the period 1975-1980 will arise. Now, from the Figure 1 it is clear, that differences in diversity and number of species before and after 1990 are not depending on the trapping effort.

Comment: Introduction: Based on what earlier knowledge did you make your hypothesis? So far it is not clear from the Introduction.

Answer: yes, directly no one author gave clues as for small mammal community changes after country regained independence. However, nearly all references in the Introduction are related to land-use changes that occurred in the country (abandonment of agricultural areas; increased land fragmentation; increased forest use), so we add text in the beginning of the sentence:

Based on small mammal responses to land-use changes [2,3,7,9,12–16], we tested the hypothesis that small mammal diversity increased after the 1990s when large scale intensive agriculture was abandoned along with the other land use changes after the regaining of country’s independence.

Some smaller remarks:

Comment: Pg 2, line 60: „Zárybnická and co-authors” number of publication missing.

Answer: [14] is the number of reference, it was present after two sentences. To make this clear, we changed text accordingly.

Comment: Pg 2, line 60-68: Here I would suggest including the effects of grassland restoration and its management on small mammals, see Mérő et al. 2015, Animal Conservation.

Answer: thank you for the reference, we have not encountered it before. In the country, we have not a grassland restoration but a loss of meadows, therefore citing this paper was done in discussion, and we added one more matching reference.

Paragraph added: The impact of land-use change on small mammal communities can vary geographically. In Hungary, for example, grassland restoration had little local impact on small mammal communities, while habitat management was important [112]. In contrast, in the mid-west region of North America, restored native grasslands have shown the importance of restored areas, acting as survival stations for voles at a time of population decline in agricultural landscapes [113]. Therefore, the synthesis of analyses of long-term changes in small mammal communities will be of great predictive value.

Round 3

Reviewer 3 Report

I can accept your responses. But I am still not convinced about keeping 2021 in the manuscript. At least indicate that 2021 is not misleading results and conclusions.